# The Role of Obesity, Body Composition, and Nutrition in COVID-19 Pandemia: A Narrative Review

**DOI:** 10.3390/nu14173493

**Published:** 2022-08-25

**Authors:** Andrea P. Rossi, Valentina Muollo, Zeno Dalla Valle, Silvia Urbani, Massimo Pellegrini, Marwan El Ghoch, Gloria Mazzali

**Affiliations:** 1Division of Geriatrics, Department of Medicine, Ospedale Cà Foncello, 31100 Treviso, Italy; 2Department of Medicine, University of Verona, 37126 Verona, Italy; 3Department of Medicine, Geriatrics Division, University of Verona, 37126 Verona, Italy; 4Department of Biomedical, Metabolic and Neural Sciences, University of Modena and Reggio Emilia, 41121 Modena, Italy; 5Department of Nutrition and Dietetics, Faculty of Health Sciences, Beirut Arab University, Beirut P.O. Box 11-5020, Lebanon

**Keywords:** SARS-CoV-2, COVID-19, obesity, body mass index, intensive care units

## Abstract

The coronavirus disease 2019 (COVID-19) pandemic has spread worldwide, infecting nearly 500 million people, with more than 6 million deaths recorded globally. Obesity leads people to be more vulnerable, developing worse outcomes that can require hospitalization in intensive care units (ICU). This review focused on the available findings that investigated the link between COVID-19, body composition, and nutritional status. Most studies showed that not only body fat quantity but also its distribution seems to play a crucial role in COVID-19 severity. Compared to the body mass index (BMI), visceral adipose tissue and intrathoracic fat are better predictors of COVID-19 severity and indicate the need for hospitalization in ICU and invasive mechanical ventilation. High volumes of epicardial adipose tissue and its thickness can cause an infection located in the myocardial tissue, thereby enhancing severe COVID-related myocardial damage with impairments in coronary flow reserve and thromboembolism. Other important components such as sarcopenia and intermuscular fat augment the vulnerability in contracting COVID-19 and increase mortality, inflammation, and muscle damage. Malnutrition is prevalent in this population, but a lack of knowledge remains regarding the beneficial effects aimed at optimizing nutritional status to limit catabolism and preserve muscle mass. Finally, with the increase in patients recovering from COVID-19, evaluation and treatment in those with Long COVID syndrome may become highly relevant.

## 1. Introduction

The coronavirus disease 2019 (COVID-19) pandemic has spread around the entire globe, infecting nearly 500 million people since the end of 2019, with more than 6 million deaths recorded globally [1]. A major proportion of these infected people have recovered with irrelevant clinical complications. However, a subgroup of individuals affected by COVID-19 developed worse outcomes leading to hospitalization in semi-intensive or intensive care units (ICU) and have shown a high rate of mortality [1].

Obesity is defined as an excessive amount of fat deposition in adipose tissue, according to the World Health Organization (WHO) [2,3]. Several studies have reported a higher prevalence of obesity in patients experiencing a severe COVID-19 clinical course, with serious complications requiring hospitalization and admission to ICU as well as a higher rate of mortality.

A meta-analysis conducted by Popkin et al. in June 2020 showed that, compared to non-obese patients, individuals with obesity have an increased risk of COVID-19 infection (over 46%), a higher risk of hospitalization (113%), a higher need for ICU admission (74%), and a risk of mortality (>48%) [4]. Obesity alone is responsible for 30% of all COVID-19 hospitalizations [5], and the rate of the latter may grow when obesity is also associated with impaired metabolic health (i.e., type 2 diabetes and hypertension) [6] even at a younger age [7,8]. Systematic reviews and meta-analyses confirmed that obesity was significantly associated with more severe forms of the disease and mortality in patients with COVID-19 [5]. These findings were also confirmed in the ICU and showed that a higher BMI is associated with higher inflammation levels, muscle damage, and in-hospital mortality during the first 28 days [9]. Moreover, being overweight in critically ill COVID-19 patients requiring invasive mechanical ventilation significantly increases their risk of death [10,11].

Even though the underlying mechanisms are still not fully understood, behind the severe prognosis of COVID-19 in individuals with obesity, increasing evidence suggests several hypotheses. Firstly, the presence of uncontrolled weight-related comorbidities (e.g., type 2 diabetes, cardiovascular, pulmonary, and renal diseases) make this population more vulnerable [12]. Pulmonary complications particularly in this population may present primary fertile soil for respiratory tract infection [13], especially lung fat embolism [14]. Secondly, due to the abnormal fat deposition, immune system alterations may facilitate a systemic diffusion of infection, and make the condition difficult to treat [15,16]. Thirdly, an increased risk of nosocomial infections [17], and the lack of full knowledge about optimum antimicrobial doses suitable for patients with obesity that fit their body weight [18] may also lead to difficulties in treating these patients in time, with possible life-threatening consequences [19].

In the field of obesity research in the last decades, body composition has gained importance as a risk factor for unfavorable health-related outcomes. Computer tomography (CT) and magnetic resonance imaging (MRI) have been established as the standard reference techniques for studies that evaluate body fat distribution (Figure 1). CT scans were considered the reference method in performing diagnoses with suspected pneumonia in symptomatic hospitalized COVID-19 patients and, as a consequence, in the last two years, many studies have investigated the relationship between thoracic and high abdominal fat distribution and several important health-related outcomes in this population. Growing available evidence on this topic should be evaluated in order to improve clinical management and better address future research.

The goals of this narrative review are (1) summarize the available findings that investigate the link between COVID-19 and body composition components, in particular different adipose tissue depots and skeletal muscle, (2) outline the potential impact of nutritional status evaluation and the implementation of specific interventions in patients with SARS-CoV-2 infection, both in general healthcare and ICU settings, and (3) identify issues not addressed by the literature and clinical implications, with particular attention paid to important outcomes in COVID-19 patients such as mortality, the risk of hospitalization, and intensive care support need.

## 2. Methods

Comprehensive research on Pubmed, Scopus, Google Scholar, and Web of Science databases was conducted in January 2022 for studies evaluating the link between COVID-19, obesity, body composition, and nutrition. The following keywords were utilized for the literature search: “acute OR post COVID-19”, “SARS-CoV-2”, “obesity OR obese population”, “adipose tissue”, “visceral adipose tissue”, “epicardial adipose tissue”, “hypercoagulability”, “sarcopenia OR sarcopenic adults”, “muscle function OR muscle quality”, “muscle loss OR damage OR atrophy”, “ immobility OR immobilization”.

The inclusion criteria were peer-reviewed original research (cross-sectional, retrospective, observational studies), review articles, and meta-analyses conducted on adults who experience COVID-19 with or without a different length of ICU admission. Published conference abstracts and editorials, as well as non-full text articles published in the English language, were excluded.

The main evidence of the relation between body composition and unfavorable COVID-19 health outcomes is summarized in Table 1.

## 3. The Role of Adipose Tissue Distribution in Patients with Severe COVID-19

### 3.1. Visceral Adipose Tissue

Epidemiological studies show that visceral adipose tissue (VAT, Figure 1, abdomen protocol, in yellow) may be a better predictor of COVID-19 severity than BMI [20,21]. Favre et al. show that for low VAT content, the linearity of the relationship is lost, at least in part due to a maldistribution of VAT values in the sample studied [21]. Interestingly, the authors showed that the correlation lost statistical significance after adjustment for sex and age, suggesting that VAT could explain the relation between sex, age, and the severity of COVID-19, which is greater in men and older populations. VAT is recognized as a risk factor for the severity and mortality of COVID-19, and multiple pathogenetic mechanisms have been proposed [22]. A previous meta-analysis identified VAT as a risk factor for hospitalization, ICU admission, and invasive mechanical ventilation [20]. Therefore, not only body fat quantity but also its distribution can play a crucial role in COVID-19 severity even though the underlying mechanisms linking central adiposity to severe COVID-19 are not completely understood. Through the possible mechanisms, inconsistent glycemic control strongly related to VAT depot [23] is associated with significantly increased mortality in COVID-19 [24]. Furthermore, during an infection, uncontrolled glycemia can lead to an immunity dysfunction that worsens patient outcomes. Individuals with obesity could undergo an additional worsening of glycemic control when dexamethasone, effective and frequently used in COVID-19 severe illness treatment, is introduced, as steroids can alter glycemic homeostasis [25].

VAT and intrathoracic fat can alter respiratory physiology by decreasing respiratory compliance. In these patients, airway resistance and respiratory work are increased. Another negative consequence is that patients with obesity use a larger fraction of oxygen uptake to support respiratory work, resulting in a decreased functional reserve. Central fat distribution has a mechanical effect, determining a reduction in lung elastic recoil, peripheral airway size, and chest wall compliance, affecting pulmonary volumes [26]. All this evidence can support the idea that visceral obesity can lead to an increased risk of respiratory failure in critically ill COVID-19 subjects.

The use of the ACE2 receptor by SARS-CoV-2 could also lead to increased neutrophil recruitment, capillary permeability, and pulmonary edema, as found by previous studies on the SARS-CoV-1 virus, which uses the same receptor to enter cells [27]. In particular, VAT presents high angiotensin converting enzyme (ACE) 2 expression. For this reason, it has been hypothesized that, as with other diseases (i.e., influenza A, HIV, cytomegalovirus), SARS-CoV-2 could use adipose tissue as a reservoir.

### 3.2. Epicardial Adipose Tissue

Epicardial adipose tissue (EAT, Figure 1, thorax protocol, in yellow) surrounds the heart between the myocardium and the visceral pericardium. It shares capillaries with the myocardium and it is not divided from it by any fascia [28]. EAT has heterogeneous cellularity, being composed mainly of adipocytes, which are smaller than those found in other body districts, particularly the subcutaneous regions, but also of stromal, vascular, and inflammatory cells [29]. EAT is involved in several functions, but the most important one is believed to be energetic activity. In fact, EAT has an increased capacity to release free fatty acids into the blood flow compared to other adipose tissues, and it has a lower glucose consumption [28].

Adipose tissue can be divided from embryologic, histologic, and functional points of view into two major groups: White and brown adipose tissue (WAT and BAT, respectively). Subcutaneous adipose tissue is formed principally by WAT, which has relatively few mitochondria and a single big lipid droplet, whilst BAT has multiple smaller lipid droplets and abundant mitochondria [30].

EAT expresses thermogenic genes associated with BAT and beige adipose tissue that can release many mediators: Tumor necrosis factor alpha (TNF-α), IL 1β, IL-1 receptor antagonist, IL-6, IL-8, IL-10, C-reactive protein (CRP), plasminogen activator inhibitor 1 (PAI1), D2 prostaglandin, and many others. The way in which those mediators could interact with the organism remains unclear, but different mechanisms have been proposed. It has been hypothesized that EAT may communicate with cardiac tissue through paracrine and vasocrine matter, as they share the same capillary circulation [30].

The volume and characteristics of EAT have been associated with inflammatory biomarkers such as CRP, IL-6, PAI-1, adiponectin, and D-dimer [31]. The thickness of EAT, measured by cardiac echography, has been associated with impaired coronary flow reserve [32] and thromboembolism, particularly with the prevalence of deep vein thrombosis [33].

COVID-19 is often associated with cardiac injury [34] and EAT, which is characterized by an elevated expression of ACE2 that could play a pathogenetic role. It has been hypothesized that patients with high volumes of EAT may present an exaggerated response to infection located in myocardial tissue and hence may experience severe COVID-related myocardial damage more frequently [34]. According to other authors, the infection may cause a reduction in ACE2 expression, resulting in a rapid worsening of cardiac function [35]. Both mechanisms are likely involved, since the reduction in ACE2 is associated with the development of an environment with higher levels of inflammatory markers, as well as with the worsening of cardiac function [36]. Particularly, ACE2, and consequently. a deficit of angiotensin (Ang) 1–7 (a peptide that binds the receptor Mas, produced by ACE2 following its binding to angiotensin II and its consequent cleavage) leads to the pro-inflammatory polarization of macrophages, resulting in the dysregulation of the inflammatory response, which has been typically observed in COVID-19. Recently, it has been reported that ICU-admitted COVID-19 patients with a higher EAT volume are at higher risk of developing pulmonary embolism, compared to those whose EAT volume is lower [37]. VAT depots, including EAT, are characterized by systemic oxidative stress, leading to the loss of the antithrombotic properties of the endothelium, increased platelet activation, and decreased fibrinolysis (Figure 2) [38]. Moreover, EAT, anatomically close to the pulmonary artery, may act as a SARS-CV-2 reservoir and potentially enable the diffusion of proinflammatory cytokines into pulmonary circulation with paracrine and vasocrine consequences on lung tissue and circulation [39]. Viral infection and proinflammatory cytokine production from VAT depots synergistically contribute to vascular endothelium, platelets, and other circulating vascular cells stimulation, thereby promoting the upregulation of procoagulant factors and adhesion molecules and concomitant downregulation of anticoagulant regulatory proteins, increased thrombin generation, and enhanced platelet activation [40], a series of events that lead to pulmonary embolism.

## 4. The Role of Skeletal Muscle Mass and Function in SARS-CoV-2 Infection

Skeletal muscle mass accounts for approximately 40% of body weight and is the major regulator of glucose homeostasis [41]. Moreover, this specialized tissue is involved in several essential activities, such as breathing, transmitting strength to the bones, maintaining posture, gait, and global locomotion [41]. Together with muscle mass, muscle strength has also emerged as a predictor of a long lifespan [42]. Higher muscle strength is associated with healthy aging and a lower risk of developing acute illness or chronic diseases, disability, hospitalization, and long-term mortality [43,44,45].

With advancing age, older adults may frequently experience sarcopenia, defined as the decline in muscle mass, muscle strength, and physical performance [46]. With sarcopenia, two different conditions, associated with a concomitant increase in frailty and muscle weakness, might occur: (*i*) Weight loss caused by a reduced appetite, characterized by a lower amount of protein intake, leading to malnutrition [47]; or (*ii*) weight gain caused by a disproportionate caloric intake due to the combination of poor physical activity and overeating, leading to sarcopenic obesity [47].

The relation between sarcopenia and COVID-19 is controversial. Some studies [48] have highlighted that the presence of sarcopenia can increase the vulnerability to COVID-19 and vice versa. The presence of sarcopenia can markedly increase infection rates, since sarcopenic patients have a poor immune response and have a predisposition to lipotoxicity, metabolic dysregulation, and inflammation [49]. Furthermore, sarcopenia is a risk factor for aspiration pneumonia caused by a loss in swallowing function, frequently observed in the oldest populations [50]. Finally, failures in respiratory muscle strength and function observed in patients with sarcopenia hamper the treatment for severe pneumonia and acute respiratory distress syndrome [49]. Hence, patients with sarcopenia are more likely to develop a severe form of COVID-19 associated with important unfavorable health outcomes.

On the other hand, severe and prolonged COVID-19 can lead to sarcopenia onset [51], since several studies [47,52] have shown skeletal muscle loss and damage in hospitalized patients. During the acute stage, the symptoms reported by patients are muscle soreness, fatigue, weaknesses, and deficits in lower extremity muscle contraction. In this scenario, older individuals with negative health conditions such as frailty, obesity, metabolic, and cardiovascular disorders are more prone to experiencing muscle impairments [48]. Aging is characterized by low-grade chronic inflammation, so-called inflammaging (71, 76). However, COVID-19-associated “cytokine storm” (i.e., an increase in IL-6, tumor necrosis factor-α, and c-reactive protein level), which enhances the inflammatory state, coupled with oxidative stress (that intensifies reactive oxygen species generation), worsens sarcopenia through a “catabolic crisis” with rapid protein degradation [53]. It is worth noting that the duration of immobility and bed rest associated with lower levels of routine physical activity could facilitate sarcopenia onset/deterioration after COVID-19 infection [50].

Prolonged immobilization reduces mechanical overloading, which acts as a stimulus for bone and muscle health homeostasis [54]. Regarding muscle mass, in a recent study, Narici and colleagues [55] summarized the impact of COVID-19-related sedentarism, reporting that even over short periods (i.e., 5 days) of bed rest, a significant reduction in quadriceps muscle mass can be observed. Muscle atrophy further deteriorated over time, with a loss of 10% and 15%, after 30 and 60 days, respectively [55], which, in physiological conditions, corresponds to a drop observed in more than 10 years [50]. After 5 days of bed rest, the loss of quadriceps strength is greater (i.e., 9%) compared to muscle mass [55].

Similar to the bed rest model, short-length severe COVID-19 hospitalization can also increase the risk of sarcopenia even without ICU admission [53]. In a cross-sectional study, the authors reported a higher prevalence of muscle weakness in upper and lower extremities (i.e., from 73% to 84% of the total sample) in those patients who were recovering from COVID-19 after a period of ~21 days in the hospital [56]. These participants presented a limited maximal voluntary contraction of both biceps and quadriceps, based on their predicted normal values. Moreover, the muscle strength of the biceps and quadriceps was negatively associated with the period of hospitalization. Finally, in this population, the patients reported reduced exercise tolerance coupled with symptoms of dyspnea and leg fatigue during several daily tasks (e.g., walking inside the hospital, eating, reaching the toilet) [56], symptoms frequently associated with sarcopenia.

Limited data exist on the effects of COVID-19 on muscle quality and function after a period of ICU. The study by Besutti et al. [57] shows the associations between general adiposity, VAT, intermuscular adipose tissue (IMAT, Figure 1, leg protocol, in purple), and intramuscular fat (mean attenuation of the pectoral muscle Figure 1, leg protocol, in green, the low attenuation muscle, and in blue, the high attenuation muscle) with clinical outcome in COVID-19 patients. In particular, total adipose tissue, VAT, and muscle density were associated with the risk of hospitalization, mechanical ventilation, or death, while for IMAT, the association with these clinical outcomes was better represented by a subdivision of the same into quartiles. Recently, Rossi et al. showed that critically ill individuals with severe COVID-19 with low-quality muscle and higher IMAT presented higher mortality and higher inflammation levels; they also experienced higher muscle damage during hospitalization as compared to subjects with low fat content inside the muscle [58].

In terms of muscle function, only one study found a reduction in muscle power after a period of hospitalization in the ICU. Despite the paucity of studies, it appears clear that COVID-19 and sarcopenia are linked to each other in a dangerous vicious cycle. Furthermore, this condition might be accelerated by aging, immobilization or bed rest, and low physical activity levels [55,59]. Future studies are needed to explore the consequences in terms of physical and muscle function after hospitalization in the ICU to target treatment, which should necessarily include a planned exercise regime as part of a multidisciplinary approach.

## 5. Nutrition in SARS-CoV-2 Prevention and Treatment

Through the modulation of immune function, nutritional status has the potential of influencing the course of viral infections, with implications for the duration, severity, and overall outcome of the disease [60,61]. It is also increasingly evident that the relationship between nutritional status and SARS-CoV-2 infection is bidirectional. Indeed, if poor nutritional status increases the risk of infection and negatively affects its course [62,63], on the other hand, the infection itself is a risk factor for worsening nutritional status [64]. Malnutrition is widely prevalent among SARS-CoV-2 patients at the time of hospitalization, indicating an increased risk of infection in malnourished patients [65]; moreover, poor nutritional status (in both under and overnutrition patients) has a negative impact on the course of the SARS-CoV-2 disease, with a growing body of evidence of increased severity, need for invasive treatment, and mortality [65,66,67,68]. The evaluation of nutritional status and the implementation of nutritional interventions are gaining a leading role in the approach to patients with SARS-CoV-2 infection, both in general healthcare and ICU settings.

### 5.1. Nutritional Prevention

Regarding prevention, although it is known that nutritional interventions can act as immunostimulators helping to prevent viral infections, it should be emphasized that data from randomized clinical trials are still lacking [69]. During the pandemic, the WHO confirmed the indications for proper nutrition, based on the guidelines already known, which recommend a Mediterranean diet, with a prevalent consumption of fresh and unprocessed foods, along with vegetables, in which the use of sugars and saturated fats and an excessive amount of salt are not recommended [70]. Furthermore, part of the general nutritional approach for the prevention of viral infections is the supplementation or at least adequate intake of vitamins and micronutrients, to maximize natural antiviral defenses; micronutrients such as vitamin A, vitamin E, cyanocobalamin, vitamin B 6, zinc, and selenium have different roles in supporting the functions of mucosal immunity and the integrity of the epithelial barrier and enhancing adaptive and innate immune functions [71]. As with other respiratory tract infections, a link between vitamin D deficiency and SARS-CoV-2 infection is emerging, as observational studies reported an association between low levels of 25-OH vitamin D and susceptibility to lower respiratory tract infections [72,73,74]. As pointed out by Mechanick et al., several fundamental gaps in the evidence remain [69], as to whether specific foods, macronutrients, or micronutrients can reduce the overall risk for SARS-CoV-2 infection or severity. As suggested by international guidelines, as a prevention strategy in the overall population and in those at risk of severe disease (e.g., the elderly and patients with altered nutritional status), it is reasonable to optimize the nutritional status, ensuring adequate caloric, protein, and micronutrients intake [75].

### 5.2. Nutritional Risk Assessment

It is increasingly apparent that nutritional care, including the identification of nutritional risk and the use of nutrition support, should be a fundamental part of management in SARS-CoV-2 inpatients. These subjects present a high nutritional risk for many reasons, including a high prevalence of comorbidities, as well as an increase in energy-protein requirements, hypercatabolism, the possible reduction in intake in the presence of gastrointestinal disorders and reduced appetite, dysphagia, dyspnea, or the need for support with invasive or non-invasive ventilation. In patients with SARS-CoV-2 infection, poor nutritional status is associated with poorer clinical outcomes [66,67,68]. Thus, the rapid assessment, identification, and treatment of poor nutritional status are essential for improving clinical outcomes in severely and critically ill SARS-CoV-2 patients. Based on the available evidence, the European Society for Parenteral and Enteral Nutrition (ESPEN) and the American Society for Parenteral and Enteral Nutrition (ASPEN) formulated recommendations regarding nutritional intervention in hospitalized patients with SARS-CoV-2 infection [75,76]. Most of the literature agrees on the need for nutritional screening with validated tools such as the Malnutrition Universal Screening Tool (MUST), Mini Nutritional Assessment (MNA), and Nutritional Risk Screening-2002 (NRS-2002) [77]. Moreover, regardless of nutritional status, it is important to identify a reduction in food intake early on, and to introduce nutritional treatment as soon as the caloric intake is less than 70% of the daily requirement. In line with previous recommendations for critically ill patients requiring the ICU who do not reach energy and protein targets through oral feeding, the importance of giving priority to enteral nutrition is emphasized, with the proposal of a parenteral route in case of specific limitations or even enteral nutrition if it is not possible to reach the target. In ventilated patients subjected to prono-supination cycles, the preferential recommendation for enteral nutrition remains valid, as the prone position per se does not represent a contraindication [78].

### 5.3. Nutritional Treatment

Based upon experts’ opinions and previous evidence, the ESPEN Guidelines recommend an energy calculation of 27–30 kcal/kg/day, protein intake up to 1.5 g/kg/day, and carbohydrates and lipids in a proportion of 70:30 and 50:50 in patients with respiratory failure. In the hyperacute phase of SARS-CoV-2 infection, the administration of low-calorie nutrition is recommended (no more than 70% of the estimated requirement) with an indication to increase the intake and reach energy targets by the fourth day. The high-protein intake recommended by ESPEN may be able to counteract protein catabolism seen in SARS-CoV-2 infection and could preserve skeletal muscle mass [79]. Additional strategies could stimulate protein anabolism. For instance, mixtures of single essential amino acids showed a direct anabolic effect and stimulated protein synthesis [80]. There is also some evidence that the use of omega-3 fatty acids could improve oxygenation during respiratory insufficiency even though stronger evidence is needed on this topic [81].

Apart from some registered clinical trials [82] strong evidence from intervention studies is lacking; experience from clinicians and guidelines related to similar disease states may serve as a foundation until more clinical data are available. The development and implementation of a personalized nutritional care plan and monitoring of response to nutritional therapy remain one of the most challenging issues in patients with SARS-CoV-2. With the rise in the number of patients recovering from previous SARS-CoV-2 infection, the evaluation of nutritional status and the formulation of indications for nutritional treatments in patients with long COVID-19 syndrome will also become extremely relevant.

## 6. Strengths and Limitations

To the best of our knowledge, this is the first narrative review highlighting the link between body composition components and unfavorable health outcomes in COVID-19 patients. However, some limitations warrant mention. Firstly, most studies evaluating body composition variables in relation to COVID-19 infection are single-center observational studies with a relatively small number of subjects, and only a few studies simultaneously evaluated different bodies.

Secondly, the standardized optimal cutoff for different body composition components is still missing and should be validated in a wider population.

Thirdly, evidence regarding the effects of COVID-19 on muscle mass is limited and, in particular, longitudinal studies evaluating muscle loss between the infection onset and recovery are still lacking. Moreover, evidence of a predisposition to COVID-19 infection in sarcopenic subjects at this point can be only hypothesized and should be confirmed by future studies.

Finally, this is a narrative and not a systematic or meta-analytic review, which could be considered a limitation, but only to a certain extent, since we still believe that it is premature to conduct a systematic review on this specific topic due to the paucity of investigations.

## 7. Clinical Implications

As the SARS-CoV-2 may continue to spread worldwide, clinicians should identify those individuals with a higher risk of unfavorable health outcomes. This revision of the literature shows that, not only obesity but both intrathoracic and intrabdominal adipose tissue depots are associated with an increased risk of severe COVID-19, hospitalization, pulmonary embolism, ICU admission, and mortality.

Despite the limited evidence in the literature, sarcopenia can be considered a risk factor for negative clinical evolutions and fatal outcomes in COVID-19 patients. On the other hand, courses of severe forms of COVID-19 are characterized by high-grade systemic inflammation levels and prolonged inactivity or bed rest, which are main risk factors for muscle quantity and quality loss. Therefore, the relation between sarcopenia and severe forms of COVID-19 can be bidirectional, triggering a vicious cycle with important consequences on autonomy levels, particularly in older adults (Figure 3). Identifying subjects with higher muscle lipid content, as evaluated with the IMAT-to-muscle ratio, should help to individuate those adults with a higher risk of muscle loss and/or damage and who could benefit from a more intensive nutritional approach to prevent or at least counteract muscle damage associated with acute phases of the disease.

Furthermore, this review shows that subjects with obesity are at a higher risk of hospitalization, which suggests that the use of BMI is an easy and quick method to stratify the risk of unfavorable outcomes in outpatient settings. For inpatients, the use of easily available CT parameters (VAT, EAT, and IMAT) in critically ill subjects undergoing standard thoracic and high-abdomen CT could help to identify, monitor, and treat subjects carefully at higher risk, in order to prevent serious life-threatening consequences and an increase in related hospital costs. In outpatient clinics, or in the absence of such measurements, the use of anthropometry, and in particular waist circumference, as a surrogate for visceral fat can be encouraged, although the available evidence on the predictive value of this parameter in severe COVID-19 is still scarce.

The identification of subjects at risk of malnutrition and the use of nutritional support should be a fundamental part of management in SARS-CoV-2 inpatients. In particular, ESPEN guidelines recommend a high protein intake, so as to counteract protein catabolism seen in SARS-CoV-2 infection, in order to preserve skeletal muscle mass. Trials investigating the effect of different nutritional approaches during severe courses of COVID-19 are ongoing, but the evidence is still lacking.

## 8. Conclusions

Both excess body fat mass, as evaluated with BMI and body fat distribution, are associated with mortality, the risk of hospitalization, and intensive care support need. Low muscle mass and muscle quality are both associated with unfavorable outcomes, particularly in critically ill elderly subjects in intensive care settings.

This review highlights the importance of body composition evaluations that are easily available in critical care settings as a clinical tool useful to identify subjects with disadvantaged body composition profiles, who have a higher risk of COVID-19-related complications and unfavorable courses.

## Figures and Tables

**Figure 1 nutrients-14-03493-f001:**
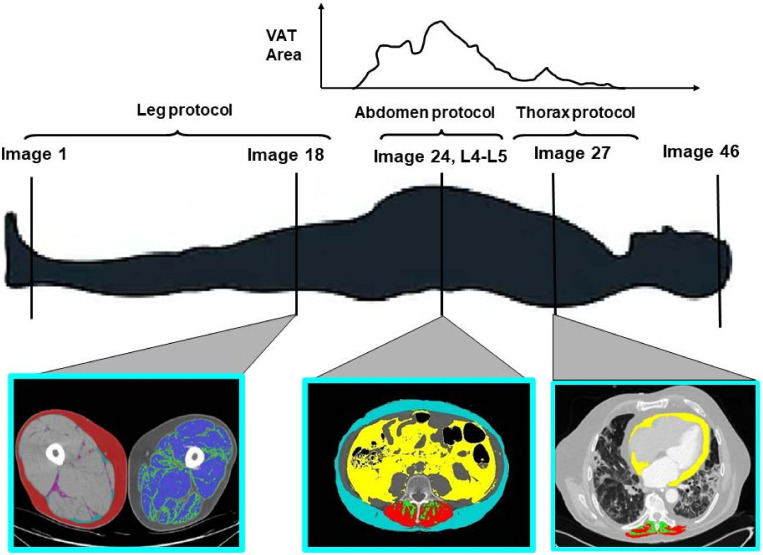
Computer tomography imaging protocol for both whole-body and regional (i.e., left to right, leg, abdomen, thorax, etc.) measures of adipose tissue and lean tissue.

**Figure 2 nutrients-14-03493-f002:**
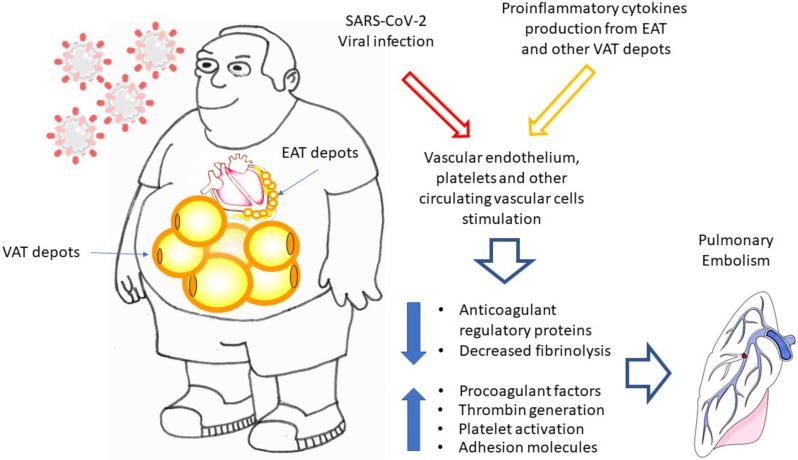
Hypothesis of the pathogenic role of epicardial and visceral adipose tissue on pulmonary embolism in COVID-19 patients.

**Figure 3 nutrients-14-03493-f003:**
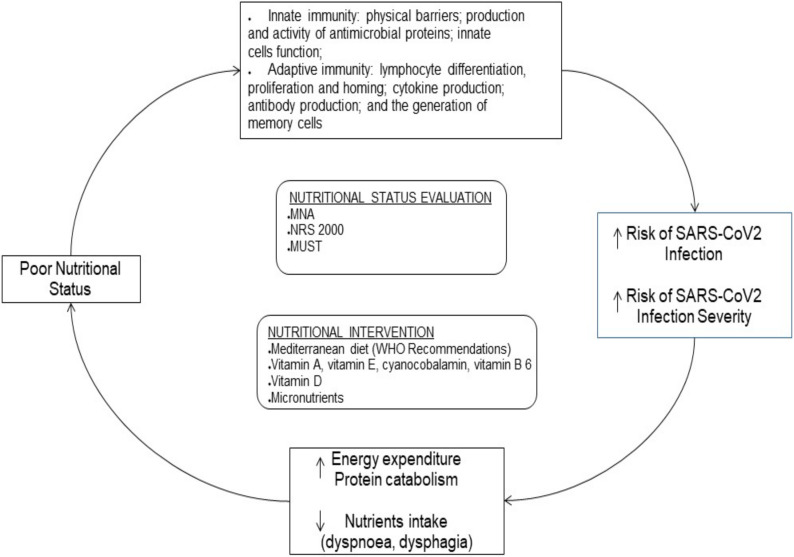
Relations between nutrition and COVID-19 infection. ↑ = increase, ↓ = decrease.

**Table 1 nutrients-14-03493-t001:** Studies investigating the relationship between different body composition components and clinical outcomes included in the narrative review.

Study (Year)	PMID	Country	Diagnostic Criteria (Parameters)	Age (Mean +/− SD)	Study Design	Sample Size and Sex Distribution (Female %)	Methods for Diagnosis	Outcome
Rossi et al. (2022)	in press N/A	Italy	Epicardial AT	64.7 (10.6)	cross-sectional	138 (45%)	Computer tomography (CT)	Association with mortality, association with pulmonary embolism
Popkin et al. (2020)	32845580	worldwide	Obesity	NA (NA)	meta-analyses	sample size varied based upon the studied outcome	Anthropometry (BMI)	Association with SARS-CoV-2 incidence, hospitalization, ICU admission, mortality
Ho et al. (2020)	33463658	worldwide	Obesity	NA (NA)	meta-analyses	sample size varied based upon the studied outcome	Anthropometry (BMI)	Obesity is associated with risk of severe disease, mortality and infection with COVID-19. Higher BMI is associated with ICU admission and critical disease
O’Hearn et al. (2021)	33629868	US	Obesity	47 (NA)	cross-sectional	11,268 (51.8%)	Anthropometry (BMI)	Obesity increases risk of hospitalization in COVID-19 by 30.2- fold
Williamson et al. (2020)	32640463	UK	Obesity	NA (NA)	cross-sectional	10,926 (44%)	Anthropometry (BMI)	Obesity increases risk of death in COVID-19 patients
Onder et al. (2020)	32812383	Italy	Obesity	70.2 (12)	cross-sectional	3694 (NA)	Anamnestic	Association with non-respiratory deaths, particularly AKI and shock
Huang et al. (2020)	33002478	worldwide	Obesity, VAT	NA (NA)	meta-analyses	NA (NA)	Anthropometry (BMI), NA for VAT	Association with hospitalization, ICU admission, IMV requirement and death. Excessive VAT is associated with severe COVID-19 outcomes.
Rossi et al. (2021)	33549439	Italy	Obesity	NA (NA)	cross-sectional	95 (18%)	Anthropometry (BMI)	Association with mortality and muscle damage
Calleluori et al. (2022)	35082385	Italy	Obesity	65 (14)	cross-sectional	42 (35%)	Anthropometry (BMI)	Fat embolism syndrome was more prevalent among COVID-19+ whether they were obese or not, fat embolism was prevalent among obese patients whether they were COVID-19+ or not. All infected subjects’ lungs presented lipids-rich hyaline membranes
Favre et al. (2021)	33246009	France	VAT	64 (17)	cross-sectional	112 (40%)	Computer tomography (CT)	Subcutaneous/visceral fat ratio was lower in patients with severe COVID-19. VAT area ≥ 128.5 cm^2^ is the best predictor for severe COVID-19. VAT was a better predictor of COVID-19 severity than BMI.
Petersen et al. (2020)	32673651	Germany	VAT	66 (13)	cross-sectional	30 (40%)	Computer tomography (CT) and abdominal circumference	VAT, both CT-measured and circumference-based, is associated with higher ICU admission and mechanical ventilation need
Menozzi et al. (2021)	35063210	Italy	Sarcopenia	71 (NA)	retrospective	272 (37%)	Computer tomography (CT)	Significant association between sarcopenia and poor clinical outcomes only during first wave
Besutti et al. (2021)	33989341	Italy	SAT, VAT, IMAT, pectoral muscle area and density	66 (NA)	observational	318 (38%)	Computer tomography (CT)	VAT and IMAT were significantly associated with hospitalization and MV or death, increased muscle density showed a protective effect on hospitalization and MV or death.
Rossi et al. (2021)	34025446	Italy	IMAT/muscle	64 (10)	cross-sectional	153 (31%)	Computer tomography (CT)	IMAT/muscle was associated with death and muscle damage in severe ICU-admitted COVID-19 patients
Foldi (2021)	33263191	worldwide	VAT, SAT	NA (NA)	meta-analyses	509 (NA)	Computer tomography (CT)	Visceral fat is associated with severity of COVID-19
Simonnet et al. (2020)	32271993	France	Obesity	60 (NA)	cross-sectional	124 (27%)	Anthropometry (BMI)	High frequency of obesity among patients admitted in ICU for SARS-CoV-2. BMI associated with IMV need.

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
