# Peer review of "The Role of Obesity, Body Composition, and Nutrition in COVID-19 Pandemia: A Narrative Review"

_nutrients, 2022, doi:10.3390/nu14173493_

Round 1

Reviewer 1 Report

The review from Rossi et al. summarizes the role of body composition and nutrition for COVID-19. The manuscript is well written, structured and informative. The studies used for the review represent data from several different countries and areas of the world, strengthening the impact of the review.

Besides there are several comments:

major comments:

- the manuscript has only 2 figures and 1 table. I would strongly recommend to illustrate the findings and assumed mechanisms with at least one more figure. As nutrition and COVID is one major point of the review and also focus of the journal I would suggest a figure 3 summarizing and illustrating the part "Nutrition in SARS-CoV2 Prevention and Treatment". E.g. showing the bidirectional relationship of nutrition and COVID infection, the treatment recommendations from ESPEN and ASPEN, the potential impact of micronutrients and Vit D. This would increase the quality and readability of the manuscript.  

- Figure legends for figure 1 and figure 2 missing. I would recommend adding those to give a better understanding of the figures. At the moneys figures seem isolated without context and are in parts hard to understand (especially figure1)

- Table  1 is missing a headline. I recommend some explanatory words regarding the table content in the headline

minor comments:

- "Patients that suffer from severe COVID-19 also show elevated levels of in-flammatory markers such as IL-1, IL-6, PCR (Page 6)". Should this be CRP instead of PCR?

- some english editing and spell check is recommended

Author Response

#Reviewer 1

Comments and Suggestions for Authors

The review from Rossi et al. summarizes the role of body composition and nutrition for COVID-19. The manuscript is well written, structured and informative. The studies used for the review represent data from several different countries and areas of the world, strengthening the impact of the review.

We thank the Reviewer for positive feedback received.

Besides there are several comments:

major comments:

- the manuscript has only 2 figures and 1 table. I would strongly recommend to illustrate the findings and assumed mechanisms with at least one more figure. As nutrition and COVID is one major point of the review and also focus of the journal I would suggest a figure 3 summarizing and illustrating the part "Nutrition in SARS-CoV2 Prevention and Treatment". E.g. showing the bidirectional relationship of nutrition and COVID infection, the treatment recommendations from ESPEN and ASPEN, the potential impact of micronutrients and Vit D. This would increase the quality and readability of the manuscript. 

We appreciate the constructive suggestion raised by the Reviewer. A new Figure illustrating the relationship between nutrition and COVID-19 infection has been added in the revised version of the manuscript. Please see the new Figure 3.

- Figure legends for figure 1 and figure 2 missing. I would recommend adding those to give a better understanding of the figures. At the moneys figures seem isolated without context and are in parts hard to understand (especially figure1)

Thank you, we missed the captions. In agreement with the Reviewer comment, Figure legends were added for both Figures and Figures citations across the manuscript.

- Table 1 is missing a headline. I recommend some explanatory words regarding the table content in the headline.

We agree with the Reviewer comment. We added the headline and also a Table Legend with explanatory words. Thank you.

minor comments:

- "Patients that suffer from severe COVID-19 also show elevated levels of in-flammatory markers such as IL-1, IL-6, PCR (Page 6)". Should this be CRP instead of PCR?

Thank you, the sentence has been revised in accordance with his comment.

- some english editing and spell check is recommended

In accordance with the Reviewer comment the manuscript has been revised by a native English speaker, Prof Mark Newman from the University of Verona.

Reviewer 2 Report

  1. The introduction section is too weak. For example, background, research gaps issues.

What’s the main aim of this review?

  1. Review methods are vaguely described and the results of own analyses are not presented

  1. The IMRD (Introduction, Method, Result, Discussion) structure is not complete. The quality is not good overall.

  1. Last pargaraph in point 1

Please mention Being overweight in critically ill COVID-19 patients requiring invasive mechanical ventilation increases their risk of death significantly e.g.

Shabanpur, M., Pourmahmoudi, A., Nicolau, J., Veronese, N., Roustaei, N., Jahromi, A. J., & Hosseinikia, M. (2022). The importance of nutritional status on clinical outcomes among both ICU and Non-ICU patients with COVID-19. Clinical nutrition ESPEN, 49, 225–231.

Czapla, M.; Juárez-Vela, R.; Gea-Caballero, V.; ZieliÅ„ski, S.; ZieliÅ„ska, M. The Association between Nutritional Status and In-Hospital Mortality of COVID-19 in Critically-Ill Patients in the ICU. Nutrients 202113, 3302.

  1. Review methods are vaguely described and the results of own analyses are not presented

6.     6.  What about clinical implication? Please add

Author Response

# Reviewer 2

Comments and Suggestions for Authors

The introduction section is too weak. For example, background, research gaps issues.

The introduction has been revised in accordance with the Reviewer comment.

What’s the main aim of this review?

Review aims have been revised in accordance with Reviewer comment.

Review methods are vaguely described and the results of own analyses are not presented

In agreement with the Reviewer comment a Methods and a Strengths and Limitations section has now been included in the revised version of the manuscript.

The IMRD (Introduction, Method, Result, Discussion) structure is not complete. The quality is not good overall.

We followed MDPI Nutrients for Review Manuscript considering that this is a Narrative review and not a Structured review and metanalysis. However we included an introduction section and better clarified our approach in the Methods section.  

Last paragraph in point 1

Please mention Being overweight in critically ill COVID-19 patients requiring invasive mechanical ventilation increases their risk of death significantly e.g.

Shabanpur, M., Pourmahmoudi, A., Nicolau, J., Veronese, N., Roustaei, N., Jahromi, A. J., & Hosseinikia, M. (2022). The importance of nutritional status on clinical outcomes among both ICU and Non-ICU patients with COVID-19. Clinical nutrition ESPEN, 49, 225–231.

Czapla, M.; Juárez-Vela, R.; Gea-Caballero, V.; ZieliÅ„ski, S.; ZieliÅ„ska, M. The Association between Nutritional Status and In-Hospital Mortality of COVID-19 in Critically-Ill Patients in the ICU. Nutrients 2021, 13, 3302.

The sentence and related references have been included at Point 1 in the last paragraph in accordance with the Reviewer comment.

Review methods are vaguely described and the results of own analyses are not presented

In agreement with the Reviewer comment a Methods section has been included in the revised version of the manuscript.

  1. What about clinical implication? Please add

We completely agree that a new section including clinical implication on the relationship between body composition, nutrition and COVID-19 should be important and we added it as the fifth section of the manuscript.

Reviewer 3 Report

The authors present a review paper without specifying what type of review it is (systematic, panoramic...) or a methodology section where the criteria used for its development can be checked (e.g. PRISMA), or search equation, etc. Methodologically, there is a lack of data to evaluate the work.

Author Response

#Reviewer 3

Comments and Suggestions for Authors

The authors present a review paper without specifying what type of review it is (systematic, panoramic...) or a methodology section where the criteria used for its development can be checked (e.g. PRISMA), or search equation, etc. Methodologically, there is a lack of data to evaluate the work.

This manuscript can be classified as a narrative Review and in agreement with the Reviewer comment a Methods section has been included in the revised version of the manuscript. In particular, we specified and displayed more clearly the keywords used for the research of the studies in relation to the Boolean operators. We also specified the criteria we used for selecting the studies which were discussed in our manuscript

Reviewer 4 Report

Rossi et al. The role of body composition and nutrition in COVID-19 pandemia, review article.

This review is huge, with endless non-readable sentences with poor grammar and language structure that are difficult to be followed. In fact, it is a puzzle with added pieces of results from various studies, without any connection, logic, or meaning, repeating the same confusing info 5 or 10 times in different ways or from different authors at various places of the text. Where is "the role of body composition and nutrition in COVID-19 pandemia"? What I read is a review mainly focusing on obesity with different topics presented in a confusing way of writing leading to nowhere. Ta COVID-19 is lost somewhere and may be forgotten or need fantasy to be understandable. In the end, there is a section about nutrition in COVID-19 with the same poor writing. What is this review take-home message? The review is just unreadable. I wanted many times to close it and stop reading anymore.

The whole review should be rewritten from scratch. Should be no more than 1/3 of this manuscript. Sections should be divided by subheadings and drastically shortened to focus on the point of a specific feature in a simple understandable way, directly related to COVID-19. Should be extendedly corrected by a professional native speaking English. All irrelevant information regarding studies in obesity or whatever else, or repetitions here and there, or unconnected phrases should be deleted. A text solid, simple, and clear with a focus on each short section should be presented. The structure should follow the aim of the study as it is indicated in the title. Could not be a review about obesity when the topic is "the role of body composition and nutrition in COVID-19". In such a case (similar title), obesity should be discussed in a focused way on its role in body composition and in nutrition in 1-2 sentences or a subheading of 2-3 maximum sentences. No more and not again. The same for sarcopenia and whatever else.   

Author Response

Comments and Suggestions for Authors

Rossi et al. The role of body composition and nutrition in COVID-19 pandemia, review article.

This review is huge, with endless non-readable sentences with poor grammar and language structure that are difficult to be followed. In fact, it is a puzzle with added pieces of results from various studies, without any connection, logic, or meaning, repeating the same confusing info 5 or 10 times in different ways or from different authors at various places of the text. 

In accordance with the Reviewer comment the manuscript has been completely revised by a native English speaker, reorganized and shortened, avoiding as much as possible repetitions.

Where is "the role of body composition and nutrition in COVID-19 pandemia"? What I read is a review mainly focusing on obesity with different topics presented in a confusing way of writing leading to nowhere. Ta COVID-19 is lost somewhere and may be forgotten or need fantasy to be understandable. In the end, there is a section about nutrition in COVID-19 with the same poor writing. What is this review take-home message? The review is just unreadable. I wanted many times to close it and stop reading anymore.

We thank the Reviewer for the time spent in evaluating our work and taking into account his observation we completely revised the manuscript better clarifying the relationship between body composition and COVID-19 disease. 

We took into account the Reviewer comment that obesity can’t be considered as part of body composition, but considering that obesity is one of the major known risk factors for unfavorable health outcomes in COVID-19 patients we decided to change the title of the narrative review including obesity. In fact, most literature is focused on the relationship between obesity per se, adipose tissue and its distribution and severe COVID-19, particularly in critically ill subjects. Conversely, only few studies focused on the role of low muscle mass on the course of the disease. However, in agreement with the Reviewer comment, wesubstantially shortened the part dedicated to obesity and removed the paragraph on the link between adipose tissue, hypercoagulability and thromboembolism in severely ill COVID-19 that could be considered out of focus. The number of references has been reduced from 109 to 82 and thus the number of words from 5234 to 4660, taking into account that, on the basis of other Reviewers comments, new subsections were added.

The whole review should be rewritten from scratch. Should be no more than 1/3 of this manuscript. Sections should be divided by subheadings and drastically shortened to focus on the point of a specific feature in a simple understandable way, directly related to COVID-19. Should be extendedly corrected by a professional native speaking English. All irrelevant information regarding studies in obesity or whatever else, or repetitions here and there, or unconnected phrases should be deleted. A text solid, simple, and clear with a focus on each short section should be presented. 

The structure should follow the aim of the study as it is indicated in the title. Could not be a review about obesity when the topic is "the role of body composition and nutrition in COVID-19". In such a case (similar title), obesity should be discussed in a focused way on its role in body composition and in nutrition in 1-2 sentences or a subheading of 2-3 maximum sentences. No more and not again. The same for sarcopenia and whatever else.

We thank the Reviewer for its suggestion and, as specified in previous answers to criticisms, we tried our best to modify the manuscript in accordance to his comment, taking into account also other reviewers’ comments.”

Round 2

Reviewer 1 Report

The manuscript has been extensively revised, improving structure, readability and content. The new figure 3 illustrates and summaries the content well. English editing improved readability a lot. No further comments

Author Response

We thank the Reviewer for the time spent evaluating our manuscript.

Reviewer 2 Report

Thank you very much. 

Author Response

(The authors gave the same response as above.)

Reviewer 3 Report

The authors present a new version of the manuscript, clarifying that it is a narrative review, with a section in the methodology section according to it. Although the search equation can be improved by using some shorteners (*, ~), the script could be accepted in the current version.

Author Response

(The authors gave the same response as above.)

Reviewer 4 Report

Much improved. Only a few minor comments

1.     The goals of this narrative review might be better described, especially numbers 2 and 3.

2.     In section 3 the role of adipose tissue... VAT and EAT could be presented in subdivisions 3.1 and 3.2

3.     Section 5 Nutrition: try to subdivide it into 2 parts with 2 subheadings

4.     Page 8.143 carbohydrates and lipids with a proportion of 30:70 and 50:50 in patients with respiratory = This is wrong: should be carbohydrates and lipids with a proportion of 70:30

5.     Conclusions: The main evidence of the relation between body composition and COVID-19 unfavorable health outcomes is summarized in Table 1. = Move this sentence to the end of the methods. This is what is missing in the methods section

Author Response

Much improved. Only a few minor comments

We thank the Reviewer for positive feedback received.

  1. The goals of this narrative review might be better described, especially numbers 2 and 3.

The manuscript has been revised accordingly.

  1. In section 3 the role of adipose tissue... VAT and EAT could be presented in subdivisions 3.1 and 3.2

We thank the Reviewer for its suggestion and modified the manuscript accordingly.

  1. Section 5 Nutrition: try to subdivide it into 2 parts with 2 subheadings

In accordance with the Reviewer comment we divided the section in three parts: nutritional prevention, nutritional risk assessment and nutritional treatment.

  1. Page 8.143 carbohydrates and lipids with a proportion of 30:70 and 50:50 in patients with respiratory = This is wrong: should be carbohydrates and lipids with a proportion of 70:30

We completely agree with the Reviewer and we modified the manuscript in accordance with his comment.

  1. Conclusions: The main evidence of the relation between body composition and COVID-19 unfavorable health outcomes is summarized in Table 1. = Move this sentence to the end of the methods. This is what is missing in the methods section.

We modified the manuscript in agreement with the Reviewer comment.